# Endoparasites of Red Deer (*Cervus elaphus* L.) and Roe Deer (*Capreolus capreolus* L.) in Serbian Hunting Grounds

**DOI:** 10.3390/ani14213120

**Published:** 2024-10-30

**Authors:** Nemanja M. Jovanovic, Tamas Petrović, Nenadovic Katarina, Dejan Bugarski, Zoran Stanimirovic, Milan Rajkovic, Marko Ristic, Jovan Mirceta, Tamara Ilic

**Affiliations:** 1Department of Parasitology, Faculty of Veterinary Medicine, University of Belgrade, Bul. Oslobodjenja 18, 11000 Belgrade, Serbia; mrajkovic@vet.bg.ac.rs (M.R.); tamara@vet.bg.ac.rs (T.I.); 2Scientific Veterinary Institute “Novi Sad”, Rumenacki put 20, 21113 Novi Sad, Serbia; tomy@niv.ns.ac.rs (T.P.); dejan@niv.ns.ac.rs (D.B.); 3Department of Animal Hygiene, Faculty of Veterinary Medicine, University of Belgrade, Bul. Oslobodjenja 18, 11000 Belgrade, Serbia; katarinar@vet.bg.ac.rs; 4Department of Biology, Faculty of Veterinary Medicine, University of Belgrade, Bul. Oslobodjenja 18, 11000 Belgrade, Serbia; zoran@vet.bg.ac.rs; 5Department of Animal Husbandry, Faculty of Agriculture, University of Nis, Kosanciceva 4, 37000 Krusevac, Serbia; markoristicnis@yahoo.com; 6Public Company Vojvodinašume, Preradovićeva 2, 21131 Novi Sad, Serbia; jovan.mirceta@vojvodinasume.rs

**Keywords:** red deer, roe deer, gastrointestinal parasites, intensity of infection, health care

## Abstract

Red deer and roe deer play an important ecological role in wildlife reserves. Given the direct and indirect damage caused by endoparasites in wild ruminants and their impact on animal welfare, it is necessary to diagnose parasitic infections. Fecal samples from 289 wild cervids were analyzed using parasitological methods, revealing a range of endoparasites, including protozoa, nematodes, cestodes, and trematodes. Infections were prevalent in both hunting areas, with parasitic gastroenteritis being the most common disease. Mixed infections, particularly triple infections, were more common than single infections. The findings highlight the importance of understanding parasitic infections in wild ruminants to inform health protection programs and control strategies in hunting grounds.

## 1. Introduction

Raising wild ruminants represents a very significant activity in hunting, which is why great attention is paid to health protection and the rational use of deer and roe deer populations [1]. Red deer and roe deer play an important ecological role in wildlife reserves, and venison meat is becoming increasingly popular due to its favorable fatty acid content and specific organoleptic characteristics [2]. In terms of negative impacts, they can be sources/reservoirs of endoparasites that affect the health status and welfare of domestic ruminants in free-range systems [3], as well as zoonotic agents that impact food safety [4,5,6,7,8]. Local residents and experts consider deer undesirable due to the damage they cause to forestry and agriculture, as well as the risks they pose to road safety, leading to traffic accidents [9,10,11].

In most European countries, red deer (*Cervus elaphus* L.) make a significant contribution to hunting and together with roe deer (*Capreolus capreolus* L.) are the most numerous and widespread wild ungulates [3]. In Serbia, the red deer and fallow deer are native species and together with roe deer, wild boars, and chamois are classified among the biologically and economically more valuable big game species. Roe deer are present in the plains of Vojvodina and the mountainous regions of central Serbia [12]. Their high reproductive potential and ability to successfully adapt to human activities have enabled roe deer to transform from a typical edge-of-forest species to important and very numerous wild animals in agroecosystems. There is no genetic differentiation between the populations of roe deer in the north and south of Serbia, separated by the Danube River. The observed heterogeneity in inbreeding levels arose due to non-random mating, which has been further reinforced by management practices rather than overall selective pressure [13].

In a study examining deer species mortality due to disease in Slovenia, parasitic infections were found to be the leading cause of death, accounting for 48% of cases [14]. Other causes included bacterial infections (14.8%), trauma (12.5%), and metabolic disorders (9.8%); less frequent causes were neoplasms, fungal infections, winter starvation, hernias, and lightning strikes. Parasite burdens are generally expected to increase with the density of deer populations [15], potentially harming the animals’ health [16,17]. The rise in parasite infections alongside reduced food availability might contribute to the lower slaughter weights seen in densely populated red deer areas in recent years. Additionally, there is concern that the parasites of wild ruminants could be transmitted to nearby domestic ruminants such as cattle and sheep, as well as to other deer species. To monitor the impact of climate change on cervid species and their parasites and to detect host-switching events, it is crucial to develop baseline data on the diversity and faunal structure of gastrointestinal nematodes among wild ruminants in Serbia.

Epidemiological research on trematodes in wild ruminants has been conducted in the hunting grounds of Vojvodina [18] and central Serbia [19], but a detailed qualitative and quantitative parasitological screening for expected endoparasites in these game species has not yet been carried out. The aim of the present study is to expand existing knowledge on the prevalence and intensity of parasitic infections in deer and roe deer in selected hunting grounds in Serbia. Given the impact of endoparasites on the health and reproductive status of red deer and roe deer, as well as the potential risks these infections pose to the welfare of high-trophy animals, the obtained results will enable the identification of risk factors influencing the occurrence, maintenance, and spread of parasitic diseases in wild ruminants.

## 2. Materials and Methods

### 2.1. Characteristics of Study Area and Hunting Grounds

The public company “Vojvodinašume” manages 17 hunting grounds, 16 of which are flatland areas, and one is a hilly mountainous hunting ground. It covers an area of 109,824 hectares, of which approximately 28,320 hectares (26%) consist of fenced hunting grounds, while the open hunting grounds in the wild constitute 81,504 hectares (74%) of the total hunting area. The research was conducted in hunting grounds from two localities in Vojvodina, Serbia, namely Srem District (hunting ground 1—HG1) and South Backa District (hunting ground 2—HG2) (Figure 1).

### 2.2. Sample Collection

The parasitological examination included 289 fecal samples from wild cervids, which were culled during regular hunting seasons from 2019 to 2023. A total of 162 red deer (88 and 74 samples from Srem and South Backa District hunting grounds, respectively) and 127 roe deer (70 and 57 samples from Srem and South Backa District hunting grounds, respectively) were examined.

Immediately after culling, feces were collected directly from the rectum as individual samples and labeled with data related to the species of each animal. The fecal samples were stored in a portable refrigerator at a temperature of +4 °C and transported to the laboratory of the Department of Parasitology at the Faculty of Veterinary Medicine, University of Belgrade, where coprological diagnostics were performed within 24–48 h.

### 2.3. Parasitological Methods

The fecal samples were examined using four standard parasitological methods, which were NaCl flotation, sedimentation, the Baermann method, and the modified McMaster technique. In brief, for the flotation method, approximately 5 g of the fecal sample was homogenized in a NaCl flotation solution (specific gravity 1.200) and passed through a sieve into a centrifuge tube with a cover slip. The tube was centrifuged at 1500 RPM for 5 min. Each sample was examined in duplicate. For the sedimentation method, approximately 5 g of the fecal sample was homogenized in tap water, passed through a sieve into a conical glass, and filled with water. After 15 min, the supernatant was discharged. The sedimentation process was repeated twice to clarify the sediment. For each sample, three slides were examined. For the Baermann method, 10 g of each sample was placed in a sieve with gauze in a funnel and covered with warm water. The sample was left to stand for 24 h. After that, the first 10 mL of fluid was collected in a tube and centrifuged. The supernatant was discarded and the sediment was examined.

For quantitative analyses, the modified McMaster technique was used to estimate eggs/oocysts/cysts per gram of feces (EPG, OPG, CPG, respectively). For this method, 3 g of each positive sample was homogenized in 42 mL of 33.3% zinc sulfate (ZnSO_4_), passed through a sieve into a second glass, and filled into both McMaster chambers. Total EPG, OPG, and CPG were calculated by multiplying the counted parasitic elements by 50.

From the samples that were positive for the presence of strongylid eggs, a coproculture was performed. Briefly, the jar containing feces was incubated in the dark at 26 °C–28 °C for 5–7 days. Samples were periodically checked and moistened if necessary. Infective L3 larvae were recovered using the Baermann technique (as described previously). A drop of the larval suspension was placed on a glass microscope slide and the larvae were killed with Lugol’s iodine solution. Morphological identification of the L3 larvae was carried out according to the protocol described by van Wyk and Mayhew [21].

### 2.4. Statistical Analyses

Statistical analysis was performed using GraphPad Prism software, version 7 (GraphPad, San Diego, CA, USA). The Chi-square (χ^2^) test was used to determine the statistical difference between the examined wild cervids from different hunting areas that tested positive for the presence of certain endoparasites. Both host species and hunting areas are independent variables. A 95% confidence interval was established in all tables, with statistical significance set at *p* < 0.05 and *p* < 0.001 levels.

## 3. Results

In the examined wild ruminants, the following parasites were found: protozoa *Eimeria* spp. and *Buxtonella sulcata*; gastrointestinal nematodes: *Trichuris* spp., strongylids; cestodes *Moniezia* spp.; respiratory nematodes: *Capillaria* spp., *Muellerius* spp., and *Dictyocaulus* spp.; and trematodes: *Fascioloides magna*, *Fasciola hepatica*, *Paramphistomum* spp., and *Dicrocoelium dendriticum* (Figure 2).

### 3.1. Prevalence of Endoparasites in Red Deer

The total prevalence of endoparasites in red deer from hunting ground 1 and hunting ground 2 was 89.77% (79/88) and 72.97% (54/74), respectively. The most prevalent protozoa were *B. sulcata* (32.10%), while the prevalence of coccidia was 11.73%. Among the Nematoda, the most prevalent were Strongylidae (63.58%). Pulmonary nematodes, *Dictyocaulus* spp. (22.22%) and *Capillaria* spp. (18.52%), were also identified. The giant liver fluke, *Fascioloides magna*, was the most prevalent Trematoda (33.95%). Additionally, lower prevalence rates were observed for *Paramphistomum* spp. (16.67%), *Fasciola hepatica* (12.35%), and *Dicrocoelium dendriticum* (4.94%). *Moniezia* spp. were found only in hunting ground 1, with a prevalence of 3.40%. Of the five types of diagnosed mixed infections, the most prevalent involved infections with three endoparasites (35.19%) (Table 1). Among red deer from the two hunting grounds, significant differences (*p <* 0.05, *p <* 0.001) in the prevalence of Strongylidae (72.7%), *F. hepatica* (17%), and *Paramphistomum* spp. were found, with a higher prevalence in hunting ground 1 (Table 1).

Table 2 presents the intensity of parasitic infections obtained using the McMaster technique. In HG1, infections with *Capillaria* spp., *Fasciola hepatica*, and *Paramphistomum* spp. were mostly found at a low level, while infections with *Moniezia* spp., *D. dendriticum,* and *F. magna* were at a moderate level. The protozoa *B. sulcata* and coccidia were mostly found at a high level, as well as Strongylidae (Table 2). In HG2, infections with *Capillaria* spp., *F. hepatica*, *Paramphistomum* spp., *F. magna*, and *D. dendriticum* were found at a low level, while *B. sulcata* was found at a moderate level. Infections with coccidia and Strongylidae were mostly at a moderate or high level (Table 2).

### 3.2. Prevalence of Endoparasites in Roe Deer

The total prevalence of endoparasites in roe deer was 92.85% (65/70) in HG1 and 85.96% (49/57) in HG2. The prevalence of coccidia was 55.12%, while that of *B. sulcata* was 8.66%. Among the Nematoda, Strongylidae were found in 59.84% of samples, while *Trichuris* spp. was found in 37.80% of samples. The prevalence of pulmonary Nematoda, *Müllerius* spp., was 20.47%. The most prevalent Trematoda was *Paramphistomum* spp. (18.90%). Among other flukes, the prevalence of *F. magna* was 16.53%, *F. hepatica* was 12.60%, and *D. dendriticum* was 4.72%. A higher prevalence of *Moniezia* spp. (54.33%) was confirmed (Table 3). Of the six types of detected mixed infections, triple infections were the most prevalent, at 36.22%. Significant differences (*p* < 0.05, *p* < 0.001) were found in the prevalence of Strongylidae (62.9%), *F. hepatica* (20%), and *D. dendriticum* (8.6%) in roe deer from HG1. Additionally, the prevalence of mixed infections involving four (38.6%) and five endoparasites (11.4%) was significantly higher (*p* < 0.001) in roe deer from HG1 (Table 3).

Table 4 presents the intensity of parasitic infections obtained using the McMaster technique. In HG1, infections with *B. sulcata*, *Trichuris* spp., *F. hepatica*, and *Paramphistomum* spp. were mostly at a low intensity. Infections with coccidia, Strongylidae, *Moniezia* spp., *D. dendriticum*, and *F. magna* were predominantly found at a moderate intensity level. In HG2, a low intensity of infections with *Trichuris* spp., *Moniezia* spp., *F. hepatica*, and *Paramphistomum* spp. were detected, while infections with coccidia and *F. magna* were at a moderate intensity. Strongylidae were predominantly found to be infections of moderate or high intensity.

### 3.3. Coproculture

Using the coproculture method, four species were identified (Figure 3). By isolating and identifying third-stage larvae from samples positive for intestinal strongylids, single infection with *Trichostrongylus axei* (14.65%) and co-infections with *Haemonchus contortus* and *Chabertia ovina* (30.10%), as well as with *T. axei*, *H. contortus*, and *Ch. ovina* (55.34%) were identified in red deer (Table 5). A significant difference (*p* < 0.001) was found in the single infection with *T. axei*, with a higher prevalence among red deer in HG2 (46.15%). Also, a significant difference (*p* < 0.05) in the prevalence of the co-infection with *T. axei*, *H. contortus*, and *Ch. ovina* was higher among red deer in HG1 (15.63%).

In roe deer, single infection with *Chabertia ovina* (9.09%) was diagnosed only in HG1. In both hunting grounds, co-infections with *Ch. ovina*, *H. contortus*, and *T. axei* (70.45–81.25%), as well as with *Oesophagostomum columbianum* in the form of quadruple infection (18.75–20.45%) were detected (Table 5).

## 4. Discussion

The conducted study provides valuable data on the prevalence and intensity of endoparasitic infections in wild cervids from northern Serbia (Vojvodina), which are crucial for assessing and preserving the welfare of these animals. High-grade parasitic infections in red deer and roe deer cause direct economic losses, while low-grade infections result in the development of subclinical disease, decreased production and reproductive performance, reduced appetite, and poorer condition, making these animals more susceptible to bacterial and viral infections. The overall prevalence of endoparasites in the examined red deer was 72.97–89.77%, and in roe deer, it was 85.96–92.85%. These findings are consistent with the results of studies conducted in Spain, Portugal, and Poland, where high prevalences of endoparasites in cervids were also confirmed [22,23,24]. Lower prevalences were recorded in deer in North America [25] and roe deer in Poland [23] compared to this study. A higher number of mixed infections were observed compared to monoinfections (9.88% in deer and 3.94% in roe deer). This contrasts with wild cervids in India, where monoinfections with gastrointestinal parasites (30%) were more prevalent than mixed infections (12%) [26].

### 4.1. Protozoa

Species from the genus *Eimeria* were identified in the examined red deer, with a prevalence of 10.8–12.5%. These findings are consistent with results from authors who reported the presence of coccidia in 10.9% of deer in Poland [23], 16.8% of deer in North America [25], and 1.37% of red deer in Portugal [4]. Higher prevalences of coccidiosis were found in red deer in Poland (74.6%) [27], Denmark (71.4%) [28], and India (28%) [26]. Compared to deer, the prevalence of coccidia was higher in roe deer (52.6–57.1%), which is consistent with findings in roe deer examined in Poland (52.1%) [27] and within the prevalence range (28.4–81.3%) detected in Italy [29]. Lower prevalences have been reported in Spain (29.2%) [30] and Denmark (30%) [28]. In Poland, 45.28% of roe deer showed oocysts per gram of feces levels ranging from 350 to 700 [24], consistent with our findings of moderate-intensity infections in 46.67–65% of roe deer. According to research from European countries, *Eimeria sordida*, *E. elaphi*, *E. austriaca*, *E. asymmetrica*, *E. robusta*, and *E. cervi* are the most prevalent coccidia in red deer [24], while roe deer are parasitized by *E. capreoli*, *E. rotunda*, *E. panda*, and *E. ponderosa*. High-intensity infections with these coccidia cause enteritis accompanied by persistent diarrhea, but wild cervids are not significant transmitters of these coccidia to domestic animals [1].

Buxtonellosis was identified in 29.5–35.1% of red deer, whereas the prevalence in roe deer ranged from 7 to 10%. These results align with the findings of Tomczuk et al. [24], who reported a prevalence of 9.4% in roe deer in Poland, although the infection intensity differed, with OPG values between 250 and 800 in our study. Intestinal ciliates diagnosed in ruminants have usually been identified as *Balantidium coli* [31] or *Buxtonella sulcata* [32]. However, it has been proposed that the species parasitizing ruminants (cattle and buffalo) should be identified as *Buxtonella sulcata*. This approach is supported by results obtained in domestic and wild ruminants [24,32,33,34,35] and the opinion of Ponce-Gordo et al. [36], who stated that it is a common mistake to identify any ciliates in the feces of animals as *Balantidium coli*. There are still controversies regarding the pathogenicity of *Buxtonella sulcata* and whether it is a commensal. Although a high intensity of infection has been observed to cause diarrhea in ruminants [33,34], it is not clear whether this protozoan is the primary cause of diarrhea.

### 4.2. Nematoda

Gastrointestinal strongylids have been diagnosed in 52.7–72.7% of red deer and 56.1–62.9% of roe deer, mostly as moderate-to-high-intensity infections, which is consistent with the findings of Santín-Durán et al. [22]. They identified moderate levels of infection with three polymorphic groups from the family Ostertagiinae in wild cervids in central Spain, indicating a positive correlation with deer population density. Helminth burden was higher in red deer and was influenced by different feeding patterns of fallow deer.

In this study, monoinfections with *T. axei* were identified among red deer in hunting ground 2, while the majority of deer from HG1 exhibited a mixed infection of *T. axei*, *H. contortus*, and *Ch. ovina* (64.06%). Among roe deer from HG2, the most prevalent mixed infection was *Ch. ovina*, *H. contortus*, and *T. axei* (81.25%). Additionally, the species *Oe. columbianum* was found as part of a quadruple infection, with a slightly higher prevalence in HG1 (20.45%).

A high prevalence of strongylids have been reported in several studies. In Romania, *H. contortus* was the most frequently detected in deer, with prevalence ranging from 33.33% to 53.57%, followed by *N. filicollis* and *O. venulosum* at 39.28% [37]. In Poland, strongylids were dominant in roe deer with a prevalence of 58.49%, primarily consisting of Trichostrongylidae (*Ostertagia*, *Spiculopteragia*, *Haemonchus*, *Trichostrongylus*) and Molineidae (*Nematodirus*) [24]. A high prevalence was also observed in Denmark, where 95.2% of red deer and 75% of roe deer were affected [28]. Similarly, in central Spain, 99% of red deer were found to be infected with nematodes from the genera *Ostertagia*, *Spiculopteragia*, *Trichostrongylus*, *Cooperia*, and *Oesophagostomum* [38]. In Germany, strongylids in deer were detected in the abomasum (100%), small intestine (65.8%), and large intestine (97.4%) [39]. A slightly lower prevalence was found in red deer (26.5%) in North America [25] and wild cervids in India, where *Haemonchus* spp. predominated (32%), followed by hookworms (21%) and *Oesophagostomum* spp. (5%) [26].

One of the main health issues among wild ruminants is parasitic gastroenteritis caused by the species *Haemonchus contortus*, which is the most common trichostrongylid in roe deer and wild goats. Parasitizing the abomasal mucosa, it feeds on blood and can cause severe anemia, with possible cases of abomasitis and duodenitis leading to death [22].

Low-grade infections with species from the genus *Trichuris* (syn. *Trichocephalus*) were diagnosed only in roe deer, with prevalence rates ranging from 33.3% to 41.4%. Similar prevalence rates of trichuriosis have been observed in moose in Norway, with rates up to 33.3% [40,41], as well as in roe deer and red deer in Belarus, with a prevalence of 37.5% [42]. In Sweden, the prevalence in moose was 38% [43], while in Russia, roe deer have shown a prevalence of 38.9% [44]. Higher prevalence rates have been reported on the Iberian Peninsula, at 53.1% [45], and in France, where the disease was more prevalent in young animals [46]. This is consistent with our finding of differences in *Trichuris* spp. prevalence among roe deer of different ages, with the highest prevalence in individuals younger than 2 years (52.6%). There are also studies demonstrating that infection with this nematode was more common in males than in females [15,45]. The prevalence of trichuriosis in red deer and roe deer varies across different regions—from high (>80%) [47], moderate (10–31%), [48,49,50,51], to low (<2–9%) [52,53,54]—which may be influenced by pathogen identification methods, geographic location, and environmental factors (seasonality).

In contrast, *Capillaria* spp. infections were found only in red deer, with prevalence rates between 14.9% and 21.6%. Based on a coprological examination, this nematode was found in roe deer in Poland with a prevalence of 5.26% [27] and 7.54% [24], while *Capillaria* spp. were identified in 9.1% of red deer in North America [25] and 52.4% of red deer and 10% of roe deer in Denmark [28]. The prevalence found in red deer in this study falls between these extremes. The first detection of *Capillaria bovis* in the small intestines and abomasum of red deer from Bosnia and Herzegovina was reported by Stevanović et al. [55].

Among lung nematodes, only *Dictyocaulus* spp. were diagnosed in 21.6–23% of red deer, while only *Muellerius* spp. were found in 17.1–24% of roe deer. Parasitic pneumonia, caused by species of the genus *Dictyocaulus*, is a common cause of mortality in cervids. In Romania, the prevalence of *Dictyocaulus* spp. infections in red deer has been documented at 57.14–71.40% [37], while in Denmark, the prevalence was 100% in red deer and 30% in roe deer [28]. *Dictyocaulus viviparus* was detected in 18% of red deer in Canada [56]. *Dictyocaulus eckerti* was found in 80.7% of red deer in Germany [39] and in 11.32% of roe deer in Poland [24]. *Dictyocaulus capreolus* has been identified in roe deer in Sweden [57], Spain [58], Romania [37], and the Czech Republic [59]. Given the seroconversion rate observed in cattle against *D. viviparus*, there is potential to use third-stage larvae of *D. capreolus* for vaccine development to prevent dictyocaulosis in cattle [57].

The most prevalent protostrongylids among wild ruminants in Bulgaria and southeastern parts of Europe are *Muellerius capillaris*, *Neostrongylus linearis*, and *Protostrongylus rupicaprae* in wild goats [60]. In red deer, *Varestrongylus sagittatus* (with a prevalence of 27%) and *Elaphostrongylus cervi* (68%) were identified in Bulgaria [61], while *E. cervi* was found with a prevalence of 47% in Germany [39]. In the United Kingdom, Simpson and Blake [62] reported the presence of *V. capreoli* protostrongylids, which cause clinically significant infections in roe deer, leading to severe pathological changes in the lungs.

### 4.3. Cestoda

Eggs of Anoplocephalidae were diagnosed in red deer in hunting ground 1 only, with a prevalence of 3.4%, which is consistent with findings of *Moniezia benedeni* in red deer in Portugal (4.11%) [4], *Moniezia* spp. in red deer in Poland (1.4%) [23], and North America (0.34%) [25]. Our positive finding of these cestodes in 31.6–72.9% of roe deer significantly exceeds reports from other authors. *Moniezia expansa*, while specific to mouflon, also parasitizes red deer and wild goats. It has been identified in roe deer in western Romania (17.8%) [37] and Poland (7.54%) [24], while *Moniezia* spp. were found in 2.2% of roe deer in Poland [23].

The higher prevalence of anoplocephalidosis in roe deer compared to red deer could be explained by the fact that small ruminants are more predisposed to this cestodosis. Additionally, red deer mainly inhabit forests, whereas roe deer move more intensively in uncultivated pastures, thus being more exposed to a higher density of non-parasitic mites infected with cysticercoids. Infections of red deer and roe deer with cestodes also pose a threat to the health of domestic ruminants raised on the outskirts of forests.

Due to the high morphological similarity of *Moniezia* eggs, the species was not identified. Based on the analysis of results from other authors, it is presumed that the infections are either mono-infections with *M. expansa* or *M. benedeni*, or co-infections with both species [4,24,37]. The precise diagnosis of these cestodes is only possible through molecular methods [63]. Although the application of such methods is debatable due to cost, it is increasingly justified and economically feasible in the diagnosis of these cestodes in horses [23]. Among other Anoplocephalidae significant for wild ruminants is *Thysanosoma actinoides* [1].

### 4.4. Trematoda

The liver fluke, *Fasciola hepatica*, has been diagnosed in 6.8–17% of red deer and 3.5–20% of roe deer. In wild ruminants, *F. hepatica* in liver tissue and bile ducts causes chronic changes accompanied by signs of general intoxication and nutritional disorders. Due to the very similar developmental cycles of these trematodes and the similar ecological conditions they require for successful maintenance, the entire epizootiological area characteristic of *F. hepatica* must be considered as a potential habitat for *Fascioloides magna* as well [64].

The giant liver fluke, *Fascioloides magna*, has been found in 29.7–37.5% of red deer and 15.8–17.1% of roe deer. In these animals, the presence of pseudocysts with adult forms of flukes has been confirmed. The giant liver fluke is distributed in the northwestern part of Serbia, in a narrow belt of floodplain forests along the Danube and Sava rivers. From there, deer migrate to Hungary and move freely in the border area between Croatia and Serbia, forming a unique epidemiological unit of significance for domestic ruminants [18]. In earlier studies conducted in Serbia on red deer in the floodplain forests of northern Serbia, a prevalence of 70.6% was found [18]. In roe deer in Croatia, a prevalence of *F. magna* of 14.97% was recorded. Autopsy findings in these deer showed the presence of pseudocysts in the liver, indicating the beginning of an adaptation process in roe deer that turns the acute and fatal disease into a chronic one. This adaptation is crucial for the survival of roe deer as aberrant hosts of this trematodosis [65].

The detection of the small liver fluke, *Dicrocoelium dendriticum*, with a prevalence of 4.5–5.4% in red deer and 8.6% in roe deer in hunting ground 1, is consistent with findings from Poland (5.66%) [24]. Higher prevalence rates of *D. dendriticum* have been reported in roe deer and deer in Romania (15%) (Hora et al., 2017). Besides *D. dendriticum*, wild ruminants can also be infected by *D. hospes* and *D. orientalis* (syn. *chinensis*), which was first identified in musk deer in the Baikal region of the former Soviet Union and has been found in some deer species in Asian countries, as well as in mouflon and deer in Europe [66].

Species from the genus *Paramphistomum* have been identified in 9.5–22.7% of red deer and 12.3–24.3% of roe deer, which is significantly higher than research results from some European countries (Finland, Germany, and the Czech Republic), where paramphistomosis in wild ruminants has been diagnosed with a prevalence of up to 11%. The clinical form of paramphistomosis is usually caused by a large number of immature migrating parasites, resulting in a low egg count in feces (50–200 EPG) or below the sensitivity threshold of the applied quantitative method (<50 EPG). The quantitative results obtained are consistent with this information, as paramphistomosis in the study was identified as low-level infections in both hunting areas. The most common trematode in the rumen and reticulum of wild ruminants is *P. cervi*, with *P. leydeni* also mentioned in the literature as a potential cause of this helminthiasis [67]. In the period between 1998 and 2004 in central Serbia, the first finding of *P. microbothrium* was histologically identified in the intestines of 52.94% of roe deer [19].

## 5. Conclusions

A high prevalence of endoparasites in red deer and roe deer in hunting grounds of North Serbia (Vojvodina) indicates the existence of numerous factors influencing the occurrence, maintenance, and spread of parasitic diseases in these game species. Understanding the etiology and epidemiology of parasitic infections allows for monitoring the health status and conservation of wildlife and is essential for improving hunting, developing hunting tourism, devising more effective forest management measures, and implementing animal and public health protection programs. Additional studies on the parasitofauna of wild ruminants and the pathogenicity of diagnosed parasites are necessary for planning parasite control strategies, aiming to reduce their infectivity and limit their spread.

## Figures and Tables

**Figure 1 animals-14-03120-f001:**
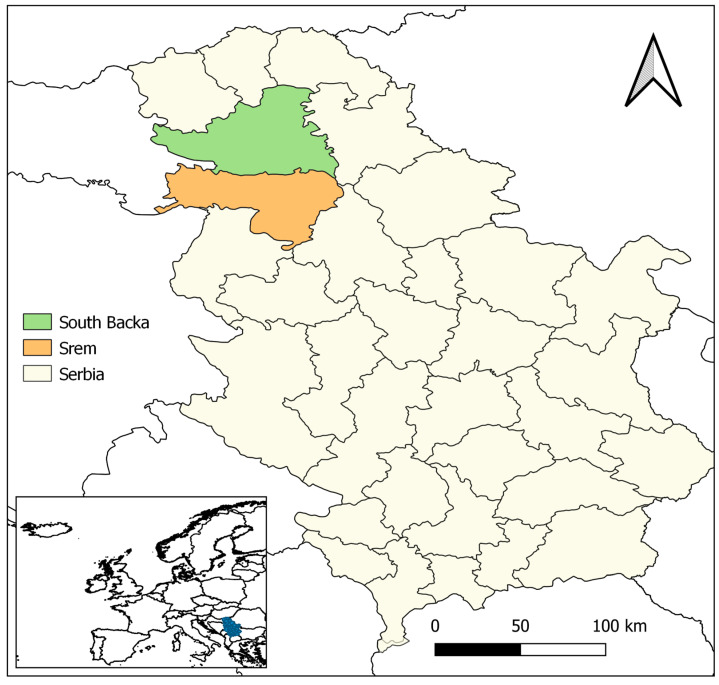
Map of Serbia with administrative districts where the survey was conducted. The map was generated by using QGIS v3.36 [20].

**Figure 2 animals-14-03120-f002:**
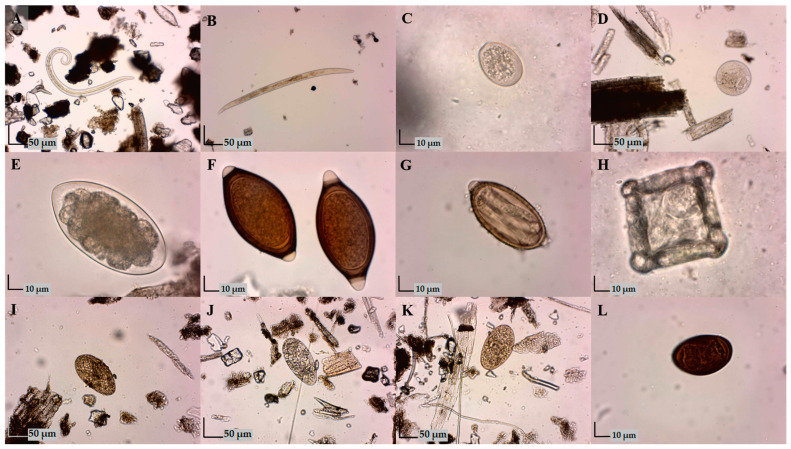
Parasitic elements detected in fecal samples: (**A**) *Müellerius* spp. (100×); (**B**) *Dictyocaulus* spp. (100×); (**C**) coccidia oocyst (400×); (**D**) *Buxtonella sulcata* cyst (100×); (**E**) Strongylidae egg (400×); (**F**) *Trichuris* spp. eggs (400×); (**G**) *Capillaria* spp. egg (400×); (**H**) *Moniezia* spp. egg (400×); (**I**) *Fasciola hepatica* egg (100×); (**J**) *Paramphistomum* spp. egg (100×); (**K**) *Fascioloides magna* egg (100×); (**L**) *Dicrocoelium dendrticum* egg (400×).

**Figure 3 animals-14-03120-f003:**
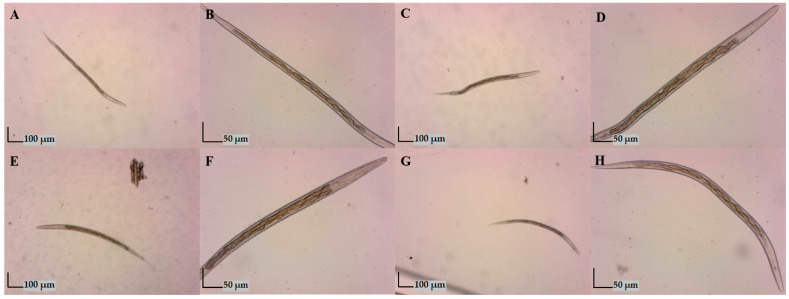
The third-stage larvae (L3) recovered using the corpoculture method (40×, 100×). (**A**,**B**) *Haemonchus contortus*; (**C**,**D**) *Chabertia ovina*; (**E**,**F**) *Oesophagostomum columbianum*; (**G**,**H**) *Trichostrongylus axei*. Morphological identification was performed according to total length, esophagus length, tail sheath length, and the number of intestinal cells [21].

**Table 1 animals-14-03120-t001:** Prevalence of endoparasites and mixed infections in red deer in selected hunting grounds.

	Hunting Ground 1		Hunting Ground 2		Total Prevalence		χ^2^	*p*
*n*	88		74		162			
Endoparasites	*N*	% (CI 95%)	*N*	% (CI 95%)	*N*	%		
Coc	11	12.50	8	10.80	19	11.73	0.11	0.74
		(5.59–19.41)		(3.73–17.87)				
BS	26	29.5	26	35.10	52	32.10	0.58	0.55
		(19.97–39.02)		(24.23–45.97)				
Cap	19	21.60	11	14.90	30	18.52	1.21	0.27
		(8.60–30.19)		(6.79–23.01)				
Str	64	72.70	39	52.70	103	63.58	6.96	***
		(63.39–82.01)		(41.32–64.08)				
Mon	3	3.40	0	0.0	3	1.85	2.57	0.11
		(1.47–5.33)						
Dic	19	21.60	17	23.00	36	22.22	0.04	0.83
		(8.60–30.19)		(13.41–32.59)				
FH	15	17.00	5	6.80	20	12.35	3.93	*
		(9.15–24.85)		(1.06–12.54)				
DD	4	4.50	4	5.40	8	4.94	0.06	0.80
		(0.17–8.83)		(0.25–10.55)				
Par	20	22.70	7	9.50	27	16.67	5.10	*
		(13.95–31.45)		(2.82–16.18)				
FM	33	37.50	22	29.70	55	33.95	1.08	0.30
		(27.38–47.62)		(19.29–40.11)				
Mixed infections								
with one parasite	9	10.20	7	9.50	16	9.88	0.03	0.87
		(3.88–16.52)		(2.82–16.18)				
with two parasites	24	27.30	13	17.60	37	22.84	2.15	0.14
		(18.00–36.60)		(8.92–26.28)				
with three parasites	28	31.80	29	39.20	57	35.19	0.96	0.33
		(22.07–41.53)		(28.09–50.31)				
with four parasites	13	14.80	5	6.80	18	11.11	2.62	0.11
		(7.38–22.22)		(1.06–12.54)				
with five parasites	4	4.50	0	0	4	2.47	3.45	0.06
		(0.17–8.83)						

*n*—number of examined samples; *N*—number of positive samples; CI—confidence interval; * *p* < 0.05; *** *p* < 0.001; Coc—coccidia; BS—*Buxtonella sulcata*; Cap—*Capillaria* spp.; Str—Strongylidae; Mon—*Moniezia* spp.; Dic—*Dictyocaulus* spp.; FH—*Fasciola hepatica*; DD—*Dicrocoelium dendrticum*; Par—*Paramphistomum* spp.; FM—*Fascioloides magna*.

**Table 2 animals-14-03120-t002:** Intensity of endoparasitic infections of red deer in selected hunting grounds.

Intensity of Infection (Quantitative FEC Method)			Endoparasites								
			Coc	BS	Cap	Str	Mon	FH	DD	Par	FM
HG1	*N*		11	26	19	64	3	15	4	20	33
	Low	*n*	0	7	16	19	1	13	0	13	9
		%	0	26.92	84.21	29.69	33.33	86.67	0	65	27.27
		Mean ± SD	0	128.6 ± 48.80	87.50 ± 56.27	131.6 ± 67.10	50	84.62 ± 51.58	0	107.7 ± 70.26	144.4 ± 63.46
	Moderate	*n*	5	9	3	20	2	2	3	7	17
		%	45.45	34.62	15.79	31.25	66.67	13.33	75	35	51.51
		Mean ± SD	510 ± 185.1	527.85 ± 201.7	266.7 ± 28.87	480 ± 182.4	375 ± 106.1	350	650 ± 217.9	478.6 ± 223.3	567.6 ± 153
	High	*n*	6	10	0	25	0	0	1	0	7
		%	54.54	38.46	0	39.06	0	0	25	0	21.21
		Mean ± SD	1025 ± 196.9	1040 ± 201.1	0	1090 ± 325	0	0	850	0	992.9 ± 139.7
HG2	*N*		8	26	11	39	0	5	4	7	22
	Low	*n*	0	10	11	5	0	5	4	7	12
		%	0	38.46	100	12.82	0	100	100	100	54.55
		Mean ± SD	0	135 ± 66.87	77.27 ± 51.79	110 ± 65.19	0	110 ± 65.19	137.5 ± 110.9	107.1 ± 53.45	83.33 ± 53.65
	Moderate	*n*	4	11	0	17	0	0	0	0	6
		%	50	42.31	0	43.59	0	0	0	0	27.27
		Mean ± SD	537.5 ± 149.3	454.5 ± 192.9	0	541.2 ± 187.3	0	0	0	0	541.7 ± 159.4
	High	*n*	4	5	0	17	0	0	0	0	4
		%	50	19.23	0	43.59	0	0	0	0	18.18
		Mean ± SD	1250 ± 219.8	1140 ± 350.7	0	1544 ± 469.3	0	0	0	0	1238 ± 349.7

HG1—hunting ground 1; HG2—hunting ground 2; *N*—number of positive samples; Low (50–200 opg/cpg/epg); Moderate (250–800 opg/cpg/epg); High (>800 opg/cpg/epg); Coc—coccidia; BS—*Buxtonella sulcata*; Cap—*Capillaria* spp.; Str—Strongylidae; Mon—*Moniezia* spp.; FH—*Fasciola hepatica*; DD—*Dicrocoelium dendrticum*; Par—*Paramphistomum* spp.; FM—*Fascioloides magna*.

**Table 3 animals-14-03120-t003:** Prevalence of endoparasites and mixed infections in roe deer in selected hunting grounds.

	Hunting Ground 1		Hunting Ground 2		Total Prevalence		χ^2^	*p*
*n*	70		57		127			
Endoparasites	*N*	% (CI 95%)	*N*	% (CI 95%)	*N*	%		
Coc	40	57.10	30	52.6	70	55.12	0.26	0.61
		(45.50–68.70)		(39.64–65.66)				
BS	7	10.00	4	7.0	11	8.66	0.35	0.55
		(2.97–17.03)		(0.38–13.62)				
Tri	29	41.40	19	33.3	48	37.80	0.87	0.35
		(29.86–52.94)		(21.07–45.53)				
Str	44	62.90	32	56.1	76	59.84	0.59	0.44
		(51.58–74.22)		(43.22–68.98)				
Mon	51	72.90	18	31.6	69	54.33	21.58	***
		(62.49–83.31)		(19.53–43.67)				
Müe	12	17.10	14	24.6	26	20.47	1.06	0.30
		(8.82–25.92)		(13.42–35.78)				
FH	14	20.00	2	3.5	16	12.60	7.76	***
		(10.63–29.37)		(0–8.27)				
DD	6	8.60	0	0.0	6	4.72	5.13	*
		(2.03–15.17)						
Par	17	24.30	7	12.3	24	18.90	2.95	0.09
		(14.25–34.35)		(3.77–20.83)				
FM	12	17.10	9	15.8	21	16.53	0.04	0.84
		(8.82–25.92)		(6.33–25.27)				
Mixed infections								
with one parasite	1	1.40	4	7.0	5	3.94	2.60	0.11
		(0–4.15)		(0.38–13.62)				
with two parasites	7	10.00	12	21.1	19	14.96	3.02	0.08
		(2.97–17.03)		(10.51–31.69)				
with three parasites	21	30.00	25	43.9	46	36.22	2.61	0.11
		(19.26–40.74)		(31.02–56.78)				
with four parasites	27	38.60	8	14.0	38	29.92	9.47	***
		(27.20–50.00)		(5.00–23.00)				
with five parasites	8	11.40	0	0	8	6.30	6.95	***
		(3.95–18.85)						
with six parasites	1	1.40	0	0	1	0.79	0.82	0.36
		(0–4.15)						

*n*—number of examined samples; *N*—number of positive samples; CI—confidence interval; * *p* < 0.05; *** *p* < 0.001; Coc—coccidia; BS—*Buxtonella sulcata*; Tri—*Trichuris* spp.; Str—Strongylidae; Mon—*Moniezia* spp; Müe—*Müellerius* spp.; FH—*Fasciola hepatica*; DD—*Dicrocoelium dendrticum*; Par—*Paramphistomum* spp.; FM—*Fascioloides magna*.

**Table 4 animals-14-03120-t004:** Intensity of endoparasitic infections of roe deer in selected hunting grounds.

Intensity of Infection (Quantitative FEC Method)			Endoparasites								
			Coc	BS	Tri	Str	Mon	FH	DD	Par	FM
HG1	*N*		40	7	29	44	51	14	6	17	12
	Low	*n*	0	3	23	9	25	14	0	13	4
		%	0	42.86	79.31	20.45	49.02	100	0	76.47	33.33
		Mean ± SD	0	183.3 ± 28.87	91.30 ± 51.46	155.6 ± 52.70	96 ± 61.10	78.57 ± 57.89	0	96.15 ± 55.76	125 ± 64.5
	Moderate	*n*	26	2	6	23	26	0	6	4	8
		%	65	28.57	20.69	52.27	50.98	0	100	23.53	
		Mean ± SD	598.1 ± 181.9	275 ± 35.36	258.3 ± 20.41	493.5 ± 172.7	475 ± 159.5	0	425 ± 140.50	350 ± 70.71	618.75 ± 155.7
	High	*n*	14	2	0	12	0	0	0	0	0
		%	35	28.57	0	27.27	0	0	0	0	0
		Mean ± SD	1057 ± 195	1050 ± 70.71	0	979.2 ± 123.3	0	0	0	0	0
HG2	*N*		30	4	19	32	18	2	0	7	9
	Low	*n*	7	2	19	7	14	2	0	6	2
		%	23.33	50	100	21.88	77.78	100	0	85.71	22.22
		Mean ± SD	150 ± 50	50	76.32 ± 45.24	128.6 ± 39.34	78.57 ± 50.82	175 ± 35.36	0	75 ± 41.83	50
	Moderate	*n*	14	2	0	12	4	0	0	1	5
		%	46.67	50	0	37.5	22.22	0	0	14.29	55.56
		Mean ± SD	525 ± 180.5	550 ± 141.4	0	483.3 ± 192.3	362.5 ± 131.5	0	0	350	440 ± 185.1
	High	*n*	9	0	0	13	0	0	0	0	2
		%	30	0	0	40.63	0	0	0	0	22.22
		Mean ± SD	1272 ± 327	0	0	1381 ± 428.4	0	0	0	0	1000 ± 70.71

HG1—hunting ground 1; HG2—hunting ground 2; *N*—number of positive samples; Low (50–200 opg/cpg/epg); Moderate (250–800 opg/cpg/epg); High (>800 opg/cpg/epg); Coc—coccidia; BS—*Buxtonella sulcata*; Tri—*Trichuris* spp.; Str—Strongylidae; Mon—*Moniezia* spp.; FH—*Fasciola hepatica*; DD—*Dicrocoelium dendrticum*; Par—*Paramphistomum* spp.; FM—*Fascioloides magna*.

**Table 5 animals-14-03120-t005:** Infective larvae of gastrointestinal nematodes through coproculture method in positive Strongylidae fecal samples from red deer and roe deer.

	Red dear							
	Hunting Ground 1		Hunting Ground 2		Total		χ^2^	*p*
*n*	64		39		103			
Strongylidae	*N*	%	*N*	%	*N*	%		
*TriA*	13	20.31	18	46.15	31	30.10	7.69	***
		(10.45–30.17)		(30.50–61.80)				
*HaeC + ChaO*	10	15.63	5	12.82	15	14.56	0.15	0.71
		(6.73–24.53)		(2.33–23.31)				
*TriA+ HaeC + ChaO*	41	64.06	16	41.03	57	55.34	5.20	*
		(52.28–75.82)		(25.59–56.47)				
	Roe deer							
	Hunting ground 1		Hunting ground 2		Total		χ^2^	*p*
*n*	44		32		76			
Strongylidae	*N*	%	*N*	%	*N*	%		
*ChaO*	4	9.09	0	0	4	5.26	3.07	0.08
		(0.60–17.58)						
*ChaO + TriA+ HaeC*	31	70.45	26	81.25	57	75	1.15	0.28
		(56.97–83.93)		(67.73–94.77)				
*ChaO + TriA + HaeC + OesC*	9	20.45	6	18.75	15	19.74	0.03	0.85
		(8.53–32.37)		(5.23–32.27)				

*n*—number of examined samples; *N*—number of positive samples; CI—confidence interval; * *p* < 0.05; *** *p* < 0.001; *ChaO*—*Chabertia ovina; HaeC*—*Haemonchus contortus*; *TriA*—*Trichostrongylus axei; OesC*—*Oesophagostomum columbianum.*

## Data Availability

The data presented in this study are available upon request from the corresponding author.

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
