# Peer review of "Endoparasites of Red Deer (*Cervus elaphus* L.) and Roe Deer (*Capreolus capreolus* L.) in Serbian Hunting Grounds"

_animals, 2024, doi:10.3390/ani14213120_

Round 1

Reviewer 1 Report

Comments and Suggestions for Authors

The title should be reworded, as its current wording suggests the results obtained by the authors, and this should be done in further chapters of this publication.

There are many inaccuracies in the Materials and Methods chapter. This chapter needs to be improved. I think that a detailed description of selection criteria for culling is unnecessary. It should be shortened to the minimum necessary, and the authors should also justify the appropriateness of their description in the research methods.

In paragraph 2.3. Sample collection there are many substantive problems. How long did it take to transport the shot animals from the research area to the laboratory where the samples were collected? Did the different times affect the results?

How was the age of the animals determined and for what purpose, since there is no information on this subject in relation to the results obtained in the further part of the work  (only in tables attached as "supplementary merial").

I suggest attaching the photographs presented in the results as supplementary material, as they are not scientific novelty.

The description of the results is very chaotic and hard to read. Of course, this may be due to the type of research results described in it, but I suggest simplifying it.

Comments on the Quality of English Language

The English language used in the work is not bad, but there is a lot to change and edit in the text. The authors should always use the full names Red deer and/or Roe dee in the described results, because the statement "Deer" used more than once can be misleading and unclear.

It is similar in the naming of research areas - their full names and abbreviations. In several cases HG1, HG2 and simultaneously HGI and HGII are used (see e.g. Table 4).

I suggest checking the manuscript very carefully to look for editorial and stylistic errors.

Author Response

The authors are thankful for the comments.

Reviewer 2 Report

Comments and Suggestions for Authors

GENERAL COMMENTS

This study consists of a non-invasive monitoring of endoparasites of red deer and roe deer in several Serbian hunting areas, through analysis of fecal samples.

The concept of “coinfection” is not used adequately in this work. Coinfection involves interaction betwen different parasites, and such relationships may be antagonistic for at least one of the parasites or beneficial for one or both interacting parasites (see, for instance: Pedersen & Fenton, 2007: TREE 22, 133–139.) So, in the absence of a specific analysis of the data, the correct term to use here would be co-occurrence or mixed infections.

Age estimation of both red and roe deer based on the number of antler points is biased, because the number of antler points is not correlated with age in cervids, particularly in roe deer, whose antlers show a very few points.

SPECIFIC COMMENTS

Keywords: I suggest changing degree of infection by intensity of parasitation.

Line 49: … represent a very significant   what? action? activity? strategy?

Line 52: venison meat?

Line 53: organoleptic, better than sensory?

Lines 60-61: … the most numerous and widespread wild ungulates.

Line 156: specific gravity or specific density?

Line 161: … to determine the statistical differences between the prevalence and intensity of parasitation in the examined wild cervids from different hunting areas? Are both host species and hunting area independent or explanatory variables to be used in the statistical analyses? Please, clarify this.

Lines 238-239: delete “high levels”.

Lines 354-357: delete this paragraph.

According to the journal guidelines for authors, references in the text must follow Vancouver style. Please, revise.

Comments on the Quality of English Language

Generally, the MS is well written, but a revision of the English style by a profesional or native translator could greatly improve the quality of the text.

Author Response

(The authors gave the same response as above.)

Round 2

Reviewer 2 Report

Comments and Suggestions for Authors

I thank the authors for taking into account all my suggestions regarding the original manuscript.

The manuscript has improved a lot.

I only noticed two minor errors:

Line 117: feces were collected (plural).

Line 118: of each animal (delete "the").